# Study the Antifungal and Ocular Permeation of Ketoconazole from Ophthalmic Formulations Containing Trans-Ethosomes Nanoparticles

**DOI:** 10.3390/pharmaceutics13020151

**Published:** 2021-01-24

**Authors:** Tarek A. Ahmed, Maram M. Alzahrani, Alaa Sirwi, Nabil A. Alhakamy

**Affiliations:** 1Department of Pharmaceutics, Faculty of Pharmacy, King Abdulaziz University, Jeddah 21589, Saudi Arabia; malzahrani1253@stu.kau.edu.sa (M.M.A.); nalhakamy@kau.edu.sa (N.A.A.); 2Department of Pharmaceutics and Industrial Pharmacy, Faculty of Pharmacy, Al-Azhar University, Cairo 11651, Egypt; 3Department of Pharmaceutics, College of Pharmacy, Umm Al-Qura University, Makkah 21955, Saudi Arabia; 4Department of Natural Products, Faculty of Pharmacy, King Abdulaziz University, Jeddah 21589, Saudi Arabia; asirwi@kau.edu.sa

**Keywords:** ketoconazole, trans-ethosomes, optimization, in situ gel, antifungal, ocular permeation

## Abstract

Ketoconazole (KET), a synthetic imidazole broad-spectrum antifungal agent, is characterized by its poor aqueous solubility and high molecular weight, which might hamper its corneal permeation. The aim was to develop an ophthalmic formulation loaded with optimized trans-ethosomal vesicles to enhance KET ocular permeation, antifungal activity, rapid drug drainage, and short elimination half-life. Four formulation factors affecting the vesicles’ size, zeta potential, entrapment efficiency, and flexibility of the trans-ethosomes formulations were optimized. The optimum formulation was characterized, and their morphological and antifungal activity were studied. Different ophthalmic formulations loaded with the optimized vesicles were prepared and characterized. The ocular irritation and in vivo corneal permeation were investigated. Results revealed that the drug-to-phospholipid-molar ratio, the percentage of edge activator, the percentage of ethanol, and the percentage of stearyl amine significantly affect the characteristics of the vesicles. The optimized vesicles were spherical and showed an average size of 151.34 ± 8.73 nm, a zeta potential value of +34.82 ± 2.64 mV, an entrapment efficiency of 94.97 ± 5.41%, and flexibility of 95.44 ± 4.33%. The antifungal activity of KET was significantly improved following treatment with the optimized vesicles. The developed in situ gel formulations were found to be nonirritating to the cornea. The trans-ethosomes vesicles were able to penetrate deeper into the posterior eye segment without any toxic effects. Accordingly, the in situ developed gel formulation loaded with KET trans-ethosomes vesicles represents a promising ocular delivery system for the treatment of deep fungal eye infections.

## 1. Introduction

Various microorganisms such as bacteria, amoeba, viruses, and fungi can cause eye infections. Fungal infections are mostly developed as a result of an eye injury and can cause severe damage to the eye. People who have had corneal surgery are at higher risk of fungal eye infections. Many different types of fungi such as fusarium, aspergillus, cryptococcus, histoplasma, zygomycetes, and candida may cause ocular infection. These fungi are widely spread and can live in soil, plants, indoor and outdoor environments, or even on the human skin and body mucous membranes. They can affect and cause serious damage to many parts of the eye such as the eyelid (eyelid nodules), conjunctiva (conjunctivitis), cornea (keratitis), choroid (choroiditis), retina (retinitis), the inside vitreous and/or aqueous humor (endophthalmitis), and the optic nerve (optic neuropathy) [1]. Mycotic keratitis and endophthalmitis are the major fungal prone infections. Mycotic keratitis usually occurs along with trauma to the cornea with the substance of plant origin. Keratitis, which is caused by fungi such as candida, typically occurs in patients with eye defects such as dry eye, chronic corneal ulceration, or corneal scarring. Endophthalmitis may be exogenous and endogenous. The former implies that blood-borne microbial infection has occurred as a result of metastatic spread of infection from other sites such as infected heart valves or the urinary tract. On the other hand, exogenous endophthalmitis results from a postoperative complication of lens removal, prosthetic lens implantation, or corneal transplantation [2,3,4].

Ketoconazole (KET) is a synthetic derivative of phenylpiperazine. It possesses a broad-spectrum antifungal activity and inhibits the synthesis of the fungal ergosterol by suppressing the activity of the cytochrome P450 14α-demethylase. KET belongs to the Class II biopharmaceutical classification system, i.e., it is characterized by low aqueous solubility or high lipid solubility (log *p* = 4.74) and high permeability [5]. It undergoes extensive hepatic metabolism by oxidative O-dealkylation and by aromatic hydroxylation. The United States Food and Drug Administration has indicated that oral KET pharmacotherapy may cause severe liver damage, adrenal gland problems, and many other side effects [6]. KET is a lipophilic broad-spectrum antifungal agent that has a high molecular weight of 531.44 Da. Accordingly, these characters hinder drug transport across the corneal stroma during the treatment of ophthalmic fungal infections, especially those that affect the posterior segment. Different formulation approaches have been developed to minimize KET side effects, enhance drug therapeutic activity, and sustain drug delivery. KET-loaded nanostructured lipid carriers, nano-sponges, poly(lactide-co-glycolide) nanoparticles, and lipid-polyethylene glycol-based nanoparticles have recently been reported to achieve these goals [7,8,9,10]. Utilization of these approaches is required for the effective treatment of fungal keratitis.

The term “trans-ethosomes” has been recently assigned for lipid-based nanovesicles that contain ethanol and an edge activator. Trans-ethosomes combine the advantages of ethosomes and transfersomes. The presence of these components makes trans-ethosomes possess superior advantages in drug delivery when compared with other delivery systems. Ethanol increases the fluidity of lipids and decreases the density of the lipid bilayer; as such, it enhances drug penetration through the tiny openings formed in the stratum corneum due to fluidization [11]. The presence of the edge activator in these vesicles weakens the phospholipid bilayer and makes the vesicle ultradeformable and highly elastic [12]. Accordingly, a positive impact on the therapeutic activity is expected from these drug-loaded nanovesicles. As far as we know, no drug-loaded trans-ethosomes nanovesicles have been investigated for their application in ocular drug delivery. The presence of ethanol in these nanovesicles is considered safe during ophthalmic treatment since ethanol has been widely used in ophthalmic surface surgeries such as photorefractive keratectomy or pterygium excision. It has also been reported to treat many corneal diseases such as recurrent corneal erosion and infectious keratitis [13].

Ocular in situ gel (ISG) preparations have gained much interest due to their benefits in enhancing the bioavailability and therapeutic responses that are linked with conventional ocular drug delivery systems. The ISG system does not cause blurred vision or irritation upon administration. Further, it provides extended drug residence time, accurate dosing, and extended shelf-life [14]. Conventional ophthalmic preparations are rapidly eliminated from the eyes due to the reflex high tear fluid turnover and eye dynamic. Polymers such as gellan gum, alginic acid, carbopol, and poloxamer can be utilized to develop aqueous solutions that undergo gelation when instilled into the eye. This effect results in a prolongation of drug contact time, a decrease in the frequency of dosing, and an increase in the transcorneal penetration. Ophthalmic ISG formulations are pharmaceutical preparations administered as liquid and congeal in the cul-de-sac in response to physiological conditions. This behavior results in higher drug bioavailability and a slower rate of drainage at the corneal site [14,15,16]. Sol-to-gel transformation can be triggered by temperature, pH, and/or ion activation [5,17].

In this study, KET trans-ethosomes nanovesicles were developed and loaded into different ophthalmic in situ gel (ISG) and hydrogel preparations to enhance KET antifungal activity and minimize the drug’s side effects usually encountered during its oral treatment. The antifungal activity, eye irritation, and ocular transport were studied to evaluate the effectiveness and safety of the prepared formulation.

## 2. Materials and Methods

### 2.1. Materials

KET was a kind gift from Saudi Arabian Japanese Pharmaceuticals Co. Ltd. (SAJA) (Jeddah, Saudi Arabia). L-α phosphatidylcholine (95%) (soy), with an average molecular weight of 775.037 was procured from Avanti^®^ polar lipids, Inc. (Alabaster, AL, USA). Tween^®^ 80 (polyoxyethylene sorbitan monooleate), methanol, ethanol, propylene glycol (PG), stearyl amine, and fluorescein isothiocyanate (FITC)-dextran were procured from Sigma-Aldrich (St. Louis, MO, USA). Poloxamer 407 was obtained from Xi’an Lyphar Biotech Co., Ltd. (Xi’an, China). Sodium alginate was purchased from VWR International Co., Ltd. (Poole, England). Carbopol 940 and hydroxypropyl methylcellulose (HPMC) Mwt 86,000 g/mol, viscosity 4000 cp (2% solution) were obtained from Acros Organics (Morris Plains, NJ, USA). All other chemicals and solvents were of analytical grade.

### 2.2. Development of KET Trans-Ethosomes Vesicles

#### 2.2.1. Draper–Lin Small Composite Experimental Design

Response surface methodology is a statistical tool used to explore the effect of several formulations and/or the processing factors on one or more response variables. The proper design is selected to optimize the process or the formulation. In this work, four formulation factors that affect the vesicle size (Y_1_), zeta potential value (Y_2_), the entrapment efficiency (Y_3_), and the flexibility (Y_4_) of KET-loaded trans-ethosomes were studied using the Draper–Lin small composite experimental design utilizing the StatGraphics Centurion XV version 15.2.05 software, StatPoint Technologies, Inc. (Warrenton, VA, USA). The design space included the effect of four variables, i.e., the drug-to-phospholipid molar ratio (X_1_), the percentage of edge activator of the total lipid (X_2_), the percentage of ethanol in the hydration medium (X_3_), and the percentage of stearyl amine of the total lipid (X_4_). The selection of these variables and their limits was based on our preliminary work and data available in the literature. Table 1 represents the levels of the studied factors and the desired goal for the studied responses used in the Draper–Lin small composite design. Eighteen runs were suggested, and the composition of these trans-ethosomal nanoparticles formulation is illustrated in Table 2.

#### 2.2.2. Preparations of KET Trans-Ethosomes

KET trans-ethosomes were prepared according to the thin-film hydration method previously described [18,19,20], except for slight modification. Based on the formulation’s composition illustrated in Table 2, the calculated amount of KET, L-α-phosphatidylcholine, tween 80, and stearyl amine was placed in a long-necked round-bottom flask and dissolved in a specified volume (50 mL) of methanol. A drug concentration of 0.1% w/v, based on the total trans-ethosomal formulation, was used. The mixture was subjected to homogenization using CF3 2EY water bath sonicator, (Ultra-wave Ltd., Cardiff, UK) until a homogenous dispersion was obtained. The organic solvent (methanol) was removed slowly under reduced pressure at 50 °C using Buchi Rotavapor R-200 (Buchi Labortechink AG, Flawil, Switzerland), until a thin clear lipid film was formed on the flask wall. The flask was kept overnight in a vacuum oven, model 6505, of Thermo Fisher Scientific (Oakwood Village, OH, USA) to ensure complete removal of methanol. The dried lipid film was hydrated at 40 °C using 50 mL of phosphate buffer pH 7.4 containing the specified amount of ethanol and PG (1%). The latter was added as an enhancer [21]. The prepared trans-ethosomes formulations were subjected to probe sonication under ice, using Sonics Vibra cell, VCX 750 (Sonics & Materials, Inc., Newtown, CT, USA) for about 10 min at an amplitude of 60% to achieve particle size reduction.

#### 2.2.3. Characterization of the Prepared Trans-Ethosomes Formulations

##### Particle Size, Polydispersity Index, and Zeta Potential Measurements

The average particle size, polydispersity index (PDI), and the zeta potential value for the 18 KET trans-ethosomes formulations were estimated using a Malvern Zetasizer Nano ZSP, (Malvern Panalytical Ltd., Malvern, UK). The dynamic light-scattering technique with noninvasive backscatter optics and laser doppler micro-electrophoresis were utilized for measurement of particle size and zeta potential, respectively. All sample measurements were repeated in triplicate.

##### Entrapment Efficiency (EE)

The percentage of KET successfully entrapped in the trans-ethosomes formulations was estimated indirectly. Briefly, the prepared trans-ethosomes dispersions were subjected to centrifugation at 20,000 rpm for 60 min at 4 °C using 3 K30 Sigma Laboratory centrifuge (Ostrode, Germany). The supernatant containing the free drug was separated, filtered through an Acrodisc^®^ syringe filter of 0.2 µm, and diluted with methanol. The concentration of free KET was analyzed spectrophotometrically at λ_max_ of 240 nm. The percentage EE of KET was calculated using the following equation:(1)$$\mathrm{EE}\;\left(\%\right)=\frac{\left(\mathrm{Total}\;\mathrm{amount}\;\mathrm{of}\;\mathrm{KET}\;\mathrm{used}\;-\;\mathrm{Calculated}\;\mathrm{amount}\;\mathrm{of}\;\mathrm{free}\;\mathrm{KET}\;\mathrm{in}\;\mathrm{the}\;\mathrm{supernatant}\right)}{\left(\mathrm{Total}\;\mathrm{amount}\;\mathrm{of}\;\mathrm{KET}\;\mathrm{used}\right)}\times 100$$

##### Vesicles Flexibility

The extrusion method was used to determine the flexibility of the developed KET trans-ethosomes vesicles as previously reported [21,22], except for slight modification. The prepared vesicles were extruded through a nylon membrane filter of pore size 0.1 mm under reduced pressure of 2.5 bar using a Charles Austin Dymax 14 air pump (Walton-on-the-Naze, UK). The percentage change in the size of the vesicles before and after extrusion was measured, and flexibility was calculated according to the following equation:(2)$$\mathrm{Flexibility}\;\left(\%\right)=\left[\;\frac{\mathrm{Vesicle}\;\mathrm{size}\;\mathrm{before}\;\mathrm{extrusion}-\;\mathrm{Vesicle}\;\mathrm{size}\;\mathrm{after}\;\mathrm{extrusion}}{\mathrm{Vesicle}\;\mathrm{size}\;\mathrm{before}\;\mathrm{extrusion}}\;\right]\times 100$$

#### 2.2.4. Draper–Lin Small Composite Statistical Analysis

The data obtained for the studied responses (Y_1_, Y_2_, Y_3_, and Y_4_) were introduced into the response column of the StatGraphics software and statistically analyzed to identify the significant independent factors affecting each response. A *p*-value < 0.05 was considered significant. The optimum desirability was identified, and the optimized formulation that achieved the study goal was proposed.

#### 2.2.5. Development of the Optimized Trans-Ethosomes Formulation

An optimized KET trans-ethosomes formulation that contained 1.71, 28.69, 42.75, and 23.36 of X_1_, X_2_, X_3_, and X_4_, respectively, were prepared and characterized for particle size (Y_1_), zeta potential (Y_2_), entrapment efficiency (Y_3_), and flexibility (Y_4_), as previously mentioned.

#### 2.2.6. Morphological Study of the Trans-Ethosomal Nanovesicles

Morphology of the optimized formulation was examined using transmission electron microscopy (TEM) model JEM-1230, (JEOL, Tokyo, Japan). A few drops of the formulation were placed on a carbon-coated grid and left for about 5 min to allow better adsorption of the nanovesicles on the carbon film. Excess liquid was removed using filter paper. Finally, a few drops of phosphotungstic acid (1%) were added, and the sample was examined.

### 2.3. Solid-State Physicochemical Characterization

#### 2.3.1. Differential Scanning Calorimetry (DSC)

The thermal behavior of pure KET, the studied phospholipid, stearyl amine, and the freeze-dried KET trans-ethosomes formulation were investigated using a Shimadzu DSC-TA-50 ESI apparatus (Shimadzu, Tokyo, Japan). A specified weight (2 mg) of each sample was placed in an aluminum crucible under an atmosphere of nitrogen. The prepared samples were heated at a rate of 10 °C/min in the range of 25–300 °C.

#### 2.3.2. Fourier Transform Infrared Spectroscopy (FT-IR)

The FT-IR spectrum of pure KET, phospholipid, stearyl amine, and the prepared trans-ethosomes formulation was studied using a Nicolet Is10 FT-IR spectrometer of Thermo Scientific, Inc. (Waltham, MA, USA). The spectra of the samples were recorded in the range of 4000–400 cm^−1^.

#### 2.3.3. X-Ray Powder Diffraction (XRPD)

To examine the change in the drug crystalline state after the development of the trans-ethosomes nanovesicles, the diffraction pattern of pure KET and the freeze-dried KET trans-ethosomes formulation were analyzed using a D/max 2500; Rigaku, powder X-ray diffractometer (Tokyo, Japan). The diffraction pattern of the studied samples was recorded at a scan speed of 0.5 °C/min.

### 2.4. Antifungal Activity of KET Trans-Ethosomes Nanovesicles

To study the antifungal activity of the optimized KET trans-ethosomes formulation, a standard strain of *Candida albicans ATCC 76615*. The agar well diffusion technique was utilized in this experiment, as previously reported [23]. Briefly, petri dishes of 150 mm diameter were prepared by placing 50 mL of Müller–Hinton agar containing 1 mL fungal culture (1 × 10^6^ CFU/mL) into each dish. The fungal strain was subsequently inoculated. Holes of 12 mm diameter were made and filled with 200 µL of the studied formulation. The prepared dishes were incubated for 4 h at 37 °C. The area where there is an absence of any fungal growth around the holes (inhibition zone) was measured using a caliper. For comparative study, the antifungal activity of the trans-ethosomes nanovesicles was compared to a drug suspension (positive control) and plain trans-ethosomes formulation (negative control).

### 2.5. Development of Ophthalmic Preparations Loaded with KET Trans-Ethosomes

#### 2.5.1. Preparation of In Situ Gel and Hydrogel Formulations

Different in situ gel (ISG) formulations were prepared by simple dispersion of the specified quantity of sodium alginate (1%), poloxamer 407 (16%), or carbopol 940 (1%) in a known volume of the optimized drug-loaded trans-ethosomes formulation over a magnetic stirrer. Stirring was continued until complete dispersion of the polymer and formation of the homogenous mixture without lumps or precipitate. HPMC (0.5% w/v) was added as a viscosity modifier. For comparative study, pure KET-loaded ISG formulations were prepared in deionized water. The prepared ISG formulations were stored at 4 °C until further characterization.

Ophthalmic HPMC hydrogels were also prepared by adding the specified weight of the polymer to a known volume of either the optimized KET trans-ethosomes nanovesicles or deionized water containing pure KET over a magnetic stirrer. Stirring was continued until complete dispersion of the polymer. The prepared hydrogels were stored at 4 °C until further characterization. A total of six ISG and two HPMC hydrogel formulations were prepared, and their composition is illustrated in Table 3.

#### 2.5.2. Characterization of the Prepared Ophthalmic Formulations

##### Rheological Properties

The rheological behavior of the prepared ophthalmic formulations (0.5 g) was studied by means of a Brookfield DV III ultra V6.0 RV cone and plate rheometer (Middleboro, MA, USA) using spindle #CPE40 at 25 ± 3 °C. Angular velocity was gradually increased from 0.5 to 100 rpm. Viscosity of the ISG formulations was measured (*n* = 3) before and after gelation. Gelation of the poloxamer-based ISG formulations (F1 and F2) was induced by increasing the temperature to 34 °C. For carbopol-based ISG formulations (F3 and F4), gelation was achieved by raising the pH to 7.4 by adding 0.5 M NaOH. Gelation of the alginate-based ISG formulations (F5 and F6) was done by adding 0.5 M CaOH.

##### In Vitro Release Study

The in vitro release of KET from the prepared ISG and hydrogel formulations was evaluated using the dialysis bag method, as previously reported [24]. A known quantity of each formulation (equivalent to 9 mg drug) was placed in a firmly sealed dialysis bag of Sigma-Aldrich Inc. (molecular weight cut-off 14 k Da). A definite volume of simulated tear fluid (STF; 0.67% w/v sodium chloride; 0.2% w/v sodium bicarbonate; and 0.008% w/v calcium chloride in deionized water; and the pH was adjusted to 7.4 using hydrochloric acid) was added in a ratio of 25:7 (formulation: STF) to mimic the condition in the human eye. The dialysis bag was immersed in a glass bottle holding 250 mL of phosphate buffer pH 7.4. The prepared bottles were kept in a shaking water bath, Model 1031; GFL Corporation (Burgwedel, Germany) at 32 °C and 100 rpm. Aliquots of 3 mL were taken at predetermined time intervals with an immediate replacement to maintain sink condition. The concentration of KET in these samples was determined spectrophotometrically at 240 nm against a blank of nonmedicated carbopol-based ISG formulation (control formulation). The experiment was done in triplicate.

To investigate KET release kinetics and the drug release mechanism from the prepared formulations, the obtained results for the in vitro release were fitted to different mathematical models, namely, zero [25], first [25], Higuchi [26], and Korsmeyer–Peppas [27,28].

### 2.6. In Vivo Ocular Irritation Test

The therapeutic dose of KET in dogs is 10–40 mg/kg/day, while rats may be dosed up to 100 mg/kg/day [29]. Since our formulation is intended for ophthalmic application, which is characterized by a low incidence of systemic side effects because only a small amount of the dose will pass the retinal-blood barrier, no KET systemic adverse reactions are expected from this formulation. To examine the eye irritation upon application of the studied formulation, the rabbit eye irritation test was performed according to the method previously described by Zhu et al. [30] and Ammar et al. [31]. Twelve healthy New Zealand white rabbits weighing 1.5–2.5 kg were used. The rabbits were obtained from the animal house of the Faculty of Pharmacy, KAU, Jeddah, KSA. The protocol for this study received prior approval from the Animal Ethics Committee of the Faculty of Pharmacy, King Abdulaziz University, Jeddah, Saudi Arabia (Approval date: June, 2020, Reference No 1021441). The work was conducted in adherence with the Declaration of Helsinki, the International Guiding Principle in Care and Use of Animals (DHEW production NIH 80-23), and the Standards of Laboratory Animal Care (NIH distribution #85-23, reconsidered in 1985). Before the start of the experiment, rabbits were adapted for at least two weeks in naturally controlled enclosures at 20 ± 1 °C with a 12/12-h dark/light cycle. Libitum was added to standard food and water.

Rabbits were randomly classified into three groups consisting of four rabbits each. Group I received ocular treatment of 25 µL of nonmedicated carbopol-based ISG formulation (control group). Group II administered the same volume of F3 (carbopol-based ISG formulation containing the optimized drug trans-ethosomes nanovesicles). The third group was given the same volume of F4 (carbopol-based ISG formulation containing pure KET). The studied formulations were instilled, using a micropipette, into the lower conjunctival sac. Eyes of the treated animals were observed over a period of 8 h. The degree of eye irritation was recorded following the classical Draize test [32].

### 2.7. Fluorescence Laser Microscope (FLM) Study

The transport of the prepared trans-ethosomes from the selected ISG formulation across the eye layers was investigated using a Zeiss Axio Observer D1 Inverted FLM of Carl Zeiss AG (Oberkochen, Germany). An optimized trans-ethosomes formulation loaded with fluorescence isothiocyanate (FITC), instead of KET, was prepared and loaded into carbopol-based ISG, as previously described. Also, a carbopol-based ISG formulation loaded with pure FITC was prepared as a control. A volume of 25 µL from each formulation was instilled, using a micropipette, into the lower conjunctival sac. Twelve healthy New Zealand white rabbits weighing 1.5–2.5 kg were used. The rabbits were randomly classified into two groups consisting of six rabbits each. The animals were sacrificed after 1, 2, and 4 h (*n* = 2 rabbits per each endpoint/formula). Both eyes of each animal were removed immediately. The whole eyes were kept in formalin, and a longitudinal section across each sample using microtome blades was performed. Paraffin wax samples were prepared. Fluorescence laser microscopic images were taken using 470/40 nm excitation, 495 beam splitter, and 525/50 nm emission.

## 3. Results and Discussion

### 3.1. Optimization of the Trans-Ethosomal Nanoparticles

KET trans-ethosomes nanovesicles have been developed utilizing the thin film hydration technique. The prepared nanovesicles were characterized for size, PDI, zeta potential, entrapment efficiency, and flexibility. Table 2 demonstrates the obtained results of nanovesicles characterization for the 18 prepared formulations. The particle size was in the range 220.67 ± 28.08 to 1063.33 ± 131.96 nm, and the obtained zeta potential values were between 12.87 ± 0.93 and 36.80 ± 1.04 mV. The EE was in the range 42.27 ± 1.35–94.52 ± 2.84%, and the flexibility was between 63.83 ± 0.44–96.24 ± 2.44%. PDI of the prepared vesicles was in the range of 0.463 ± 0.05 to 0.692 ± 0.05, which was an indication of nanovesicles’ homogeneity and acceptable size distribution. It has been previously reported that PDI values smaller than 0.05 were mainly observed with highly monodispersed systems, while PDI values greater than 0.7 specify a broad particle size distribution [33].

The obtained values for vesicle size, zeta potential, entrapment efficiency, and vesicle flexibility were statistically analyzed by multiple regression analysis and two-way analysis of variance (ANOVA) to identify the effect of X_1_, X_2_, X_3_, and X_4_ on Y_1_, Y_2_, Y_3_, and Y_4_. Regression analysis is a statistical tool that evaluates and analyzes the relationship between a dependent variable and one or more of the studied responses. ANOVA was used to study the main, interaction, and quadratic effect of X_1_, X_2_, X_3_, and X_4_ on Y_1_, Y_2_, Y_3_, and Y_4_. Hence, the estimated effect of factors, *F*-ratios, and *p*-values were calculated. A positive sign for the estimated effect indicated a synergistic effect of this factor on the studied response, while a negative sign denoted an antagonistic effect. The *F*-ratio was used to correlate the observed and expected averages. A value of the *F*-ratio greater than 1 indicated of a location effect; hence, the calculated *p*-value was used to specify if there is a significant level. A *p*-value that differed from zero and was less than 0.05 indicated a significant effect. Table 3 illustrated the obtained values for the estimated effect of factors, *F*-ratios, and the associated *p*-values.

#### 3.1.1. Influence of the Independent Variables (X_1_–X_4_) on the Particle Size (Y_1_)

Statistical analysis for the effect of X_1_–X_4_ on the particle size (Y_1_) indicated that the drug-to-phospholipid molar ratio (X_1_), the percentage of the edge activator of the total lipid (X_2_), and the percentage of ethanol in the hydration medium (X_3_) had a significant effect on the particle size at *p*-values of 0.037, 0.023, and 0.032, respectively. These results were also confirmed after investigation of the Pareto chart depicted in Figure 1. The polynomial equation that relates X_1_–X_4_ and Y_1_ is as follows:(Y_1_) = −3010.59 + 1374.71X_1_ + 63.370X_2_ + 10.919X_3_ + 138.493X_4_ − 77.195X_1_^2^ − 14.719X_1_X_2_ − 3.883X_1_X_3_ − 24.352X_1_X_4_ − 0.163X_2_^2^ − 0.379X_2_X_3_ − 1.898X_2_X_4_ − 0.051X_3_^2^ + 0.296X_3_X_4_ − 1.523X_4_^2^(3)

To demonstrate the effect of changing the levels of two factors on Y_1_, when the other two factors were kept at their intermediate levels, three-dimensional (3D) response surface plots were constructed and are graphically illustrated in Figure 2.

Upon increasing the drug-to-phospholipid molar ratio (X_1_) from 1:2 to 1:4, the particle size was increased. This effect could be attributed to the formation of multilamellar vesicles, as previously described by Harbi et al. [34] and Ahmed [35]. The percentage of the edge activator of the total lipid (X_2_) antagonistically affected the particle size owing to the reduction in the surface tension of the media, which leads to phospholipid arrangement in small vesicles [24,36]. The percentage of ethanol in the hydration medium also showed an antagonistic effect on the particle size, an effect that may be attributed to the formation of a phase with an interpenetrating hydrocarbon chain, which leads to a significant reduction in the membrane thickness [37]. Another possible explanation for this effect is the reduction in the main transition temperature of the phospholipids, which results in partial fluidization of the prepared vesicles and the formation of small nanoparticles upon increasing the ethanol content [38]. The obtained R^2^ value showed that the model as fitted explains 97.21% of the variability on the particle size, while the adjusted R^2^ value, which is more appropriate for comparing models with different independent variables, was 84.16%.

#### 3.1.2. Influence of the Independent Variables (X_1_–X_4_) on Zeta Potential (Y_2_)

The percentage of stearyl amine of the total lipid (X_4_) and its quadratic effect (X_4_X_4_) had a significant effect on the zeta potential (Y_2_) at *p*-values of 0.005 and 0.046, respectively. The Pareto chart shown in Figure 1 confirms this finding. The mathematical model for the zeta potential was generated, and the polynomial equation that best fit the model is: (Y_2_) = −51.045 + 10.352X_1_ + 2.198X_2_ + 0.191X_3_ + 4.429X_4_ − 0.538X_1_^2^ − 0.323X_1_X_2_ − 0.014X_1_X_3_ − 0.095X_1_X_4_ − 0.033X_2_^2^ − 0.002X_2_X_3_ + 0.001X_2_X_4_ − 0.002X_3_^2^ − 0.0005X_3_X_4_ − 0.091X_4_^2^(4)

Figure 3 shows the 3D estimated response surface plots for the effect of changing two factors on zeta potential (Y_2_) when the other two factors were kept at their intermediate levels.

All the prepared trans-ethosomes nanovesicles showed positive zeta potential values that were significantly affected by the percentage of stearyl amine of the total lipid (X_4_). As the concentration of stearyl amine was increased, the obtained zeta potential value was increased owing to further deposition of the charge-inducing agent on the vesicles’ outer surface. A previous study indicated an increase in the zeta potential value of positively charged rosuvastatin flexible liposomes upon increasing the concentration of the coating positively charged polymer (chitosan) [35]. The obtained R^2^ value showed that the model as fitted explains 98.52% of the variability on zeta potential, while the adjusted R^2^ value, which is more appropriate for comparing models with different independent variables, was 91.64%.

#### 3.1.3. Influence of the Independent Variables (X_1_–X_4_) on the Entrapment Efficiency (Y_3_)

It was noted from the ANOVA that the main effect of the drug-to-phospholipid molar ratio (X_1_) and the quadratic effect of the ethanol percentage in the hydration medium (X_3_X_3_) had a significant effect on the EE at *p*-values of 0.049 and 0.016, respectively. This finding was also confirmed after studying the Pareto chart illustrated in Figure 1. The polynomial equation that correlates the studied independent variables and the entrapment efficiency (Y_3_) is
(Y_3_) = 142.860 − 14.497X_1_ − 5.119X_2_ + 1.625X_3_ − 1.253X_4_ − 0.964X_1_^2^ + 0.778X_1_X_2_ + 0.029X_1_X_3_ − 0.207X_1_X_4_ − 0.015X_3_^2^ − 0.009X_2_X_3_ + 0.242X_2_X_4_ − 0.026X_3_^2^ − 0.002X_3_X_4_ − 0.058X_4_^2^(5)

The effect of changing two factors on the EE (Y_3_) when the other two factors were kept at their intermediate levels is shown in Figure 4.

The EE of the prepared nano-vesicles was significantly affected by the drug-to-phospholipid molar ratio (X_1_) in an antagonistic way; this effect could be attributed to the decrease in the drug load of the prepared nanovesicles upon increasing X_1_. This finding was also observed during the development of *α*-tocopherol liposomal formulation. The authors reported that the drug encapsulation efficiency was dramatically decreased when using *α*-tocopherol-to-phospholipid molar ratios of 1:10 or more [39]. In another study, the effect of changing the drug to a phospholipid molar ratio on the EE of sildenafil-loaded transferosomes was studied. The author reported an initial increase in the EE when the amount of lipid was increased; however, a further increase in the amount of lipid resulted in a decrease in the drug EE. The author attributed this finding to the competition between the drug and the phospholipid at high lipid load [40]. The obtained R^2^ value showed that the model as fitted explains 97.11% of the variability on the entrapment efficiency, while the adjusted R^2^ value was found to be 83.63%.

#### 3.1.4. Influence of the Independent Variables (X_1_–X_4_) on Flexibility (Y_4_)

ANOVA revealed a marked significant effect of X_2_ (the edge activator of the total lipid, *p*-value = 0.013) and X_3_ (the ethanol in the hydration medium, *p*-value = 0.034) on the flexibility (Y_4_). The polynomial equation of the model is
(Y_4_) = 71.769 + 1.153X_1_ − 0.926X_2_ − 0.198X_3_ + 0.107X_4_ − 0.468X_1_^2^ + 0.240X_1_X_2_ + 0.045X_1_X_3_ − 0.236X_1_X_4_ + 0.045X_2_^2^ − 0.005X_2_X_3_ − 0.011X_2_ X_4_ + 0.00003 X_3_^2^ + 0.022X_3_X_4_+ 0.009X_4_^2^(6)

The effect of the studied factors on the flexibility of the prepared nanovesicles is illustrated in the 3D response surface plots (Figure 5).

Upon increasing the edge activator (X_2_) and the ethanol concentration in the hydration medium (X_3_), the flexibility of the vesicles was increased. The effect of X_2_ may be attributed to the insertion of more surfactant molecules into the membrane of the vesicles with the lipophilic part (oleate residue) arranged parallel to the acyl chains of the phospholipid and the hydrophilic head (polyoxyethylene units) directed toward the head group of the phospholipids, which leads to the nanovesicles’ flexibility [41]. The effect of X_3_ could be related to partial fluidization of the prepared vesicles, as discussed above. Another possible explanation for the effect of ethanol concentration is the decrease in the interfacial tension of the vesicle membrane upon increasing the ethanol concentration [42]. The obtained R^2^ value showed that the model as fitted explains 97.24% of the variability on the vesicle flexibility, while the adjusted R^2^ value was 84.37%.

### 3.2. Development of the Optimized Trans-Ethosomal Formulation

Based on the multiple response statistical analysis for the obtained data, the optimum desirability that achieved the study’s goals were X_1_, X_2_, X_3_, and X_4_ levels of 1.71, 28.69, 42.76, and 23.36, respectively. An optimized KET trans-ethosomes formulation that contained these values was prepared and characterized for Y_1_–Y_4_, as previously mentioned. The optimized drug-loaded trans-ethosomes formulation showed an average size of 151.34 ± 8.73 nm, a PDI value of 0.511 ± 0.057, a zeta potential value of + 34.82 ± 2.64 mV, an EE of 94.97 ± 5.41%, and a flexibility of 95.44 ± 4.33%. The predicted values for Y_1_, Y_2_, Y_3_, and Y_4_ were 157.67 nm, +36.82 mV, 96.85%, and 96.07%, respectively.

### 3.3. Morphological Study of the Trans-Ethosomal Nanoparticles

TEM image for the optimized trans-ethosomes formulation revealed spherical-shaped vesicles, as depicted in Figure 6. The size of the vesicles displayed in the TEM image was smaller than that obtained by the dynamic light-scattering technique using the Malvern Zetasizer. This finding could be attributed to the solvent-removal effect during sample preparation for TEM imaging, which may cause size modifications. This explanation has been previously described by Das and Chaudhury, which illustrated that the sample preparation (e.g., solvent removal) might directly affect the particle size and shape [43]. The same finding was also mentioned during size determination and morphological characterization of finasteride microplates [44].

### 3.4. Solid-State Physicochemical Characterization

#### 3.4.1. Differential Scanning Calorimetry

The DSC thermogram of pure KET revealed a distinguishing sharp endothermic peak at 150.47 °C. L-α phosphatidylcholine showed a phase transition temperature at about 48 °C corresponding to the chain-melting transition. When the phospholipid sample was heated above the transition temperature, the phospholipid molecules were arranged in a multilayered liquid-crystalline phase structure, which exhibits a greater degree of motional freedom and undergoes several thermo-tropic transitions above the phospholipid phase transition temperature and below the isotropic melts (which usually occurs at about 230 °C for most phospholipids) [45]. This behavior was obvious in the DSC thermogram of the studied phospholipid (Figure 7), which exhibited a major thermo-tropic transition peak at about 190 °C. The freeze-dried optimized drug-loaded formulation showed a slight shift in the drug peak to 156.73 °C and in the phospholipid major thermo-tropic transition peak to 166.80 °C. This finding indicates the molecular dispersion of the drug in the optimized trans-ethosomes formulation; however, further investigation using another tool such as the XRPD is required to ensure crystalline transformation.

#### 3.4.2. FT-IR Spectroscopy

The FT-IR spectrum (Figure 8) of KET demonstrated a characteristic drug peak of the carbonyl group [C=O] stretching vibration at 1647.26 cm^−1^. Other characteristic drug peaks for the C–O stretching of the cyclic ether and C–O stretching of the aliphatic ether group were observed at 1244 and 1031.95 cm^−1^, respectively. The FT-IR spectrum of the studied phospholipid demonstrated a vibration band at 1200–1145 cm^−1^ due to the PO_2_ group vibration. Another band for the carbonyl group (C=O) was observed at 1765–1720 cm^−1^ [46]. The C–H bands of the phospholipid were detected at 2925 and 2855 cm^−1^ [47]. Stearyl amine displayed a maximum absorbance around 2925 and 2855 cm^−1^ due to the C–H bands. The characteristic C–N stretch of stearyl amine was detected at 1300 cm^−1^, as a sharp medium intensity peak, and the NH_2_ wagging vibration was observed at 600 cm^−1^ [47]. The spectra of the optimized trans-ethosomes formulation showed a new significant broad stretch that appeared at 3050–3500 cm^−1^, corresponding to the O–H group, which formed during the electrostatic interaction between the amino group of the stearyl amine and the carbonyl group of the phospholipid. This finding was previously mentioned by Zidan et al. during the development of tenofovir liposome formulation [47]. This electrostatic interaction was also confirmed by the disappearance of the phospholipid carbonyl group, which was observed between 1765–1720 cm^−1^. The spectra of the optimized formulation also showed a decrease in the intensity, slight shift, and some overlapping in the characteristic peaks of the drug, the phospholipid, and stearyl amine. Accordingly, formation of a lipid-based nanovesicles with a modified charged surface that encloses KET is most likely to occur.

#### 3.4.3. X-Ray Powder Diffraction

Figure 9 illustrates the XRPD patterns of pure KET and the optimized freeze-dried nanovesicles’ formulation. The pure drug demonstrated a crystalline nature as specified by the presence of numerous and sharp peaks in its diffraction spectrum. This behavior was changed in the diffraction pattern of the freeze-dried optimized nanovesicles’ formulation. There was a decrease in the intensity, disappearance, and development of some peaks, which is an indication of crystalline transformation and change in the drug nature from crystalline to amorphous.

### 3.5. Antifungal Activity of KET Trans-Ethosomes Nanoparticles

The antifungal activity of the optimized KET loaded trans-ethosomes nanovesicles (test) was compared with a drug suspension (positive control) and to a plain trans-ethosomes formulation (negative control). A great antifungal activity, as indicated by a large inhibition zone, was observed in the dish treated with the optimized KET-loaded trans-ethosomes formulation (26 mm). Small inhibition was noted in the dish treated with the drug suspension (11 mm), and no inhibition (0 mm) was observed after treatment with the plain (nonmedicated) trans-ethosomes nanovesicles. These results indicate improvement in KET antifungal activity upon loading into trans-ethosomes nanovesicles, which enhanced the drug diffusion, as previously mentioned for KET-loaded polymeric poly(lactide-co-glycolide) nanoparticles [5].

A previous study indicated that the minimum inhibitory concentration value of KET against yeast isolates obtained from cases of keratitis is 0.015–0.125 µg/mL [48]. In this study, the optimized trans-ethosomes dispersion loaded with 0.1% w/v KET, which showed an EE of 94.97 ± 5.41%, was found to be an effective antifungal preparation.

### 3.6. Characterization of Ophthalmic Preparations Loaded with KET Trans-Ethosomes

#### 3.6.1. Rheological Properties

All the studied formulations exhibited a decrease in viscosity upon increasing the shear stress, which is an indication of a pseudoplastic behavior, as demonstrated in Table 4. The ISG formulations were liquid at room temperature (25 °C) and transformed rapidly into the gel phase upon exposure to the gelation stimuli (increasing the temperature to 34 °C, raising the pH to 7.4, or adding 0.5 M CaOH). These stimuli were used to mimic the eye’s biological condition. The type of polymer used to develop the ISG formulation exhibited a significant effect on the formulation viscosity (Table 4). Higher viscosity value was observed with ISG formulations containing sodium alginate (F5–F6), followed by ISG formulations containing carbopol (F3–F4), and finally ISG formulations containing poloxamer (F1–F2). This finding is in good agreement with a previous work of Basaran and Bozkir, who developed ophthalmic ISG formulations of ciprofloxacin hydrochloride using poloxamer and carbopol polymers [16]. Unlike the ISG, hydrogels are simple viscous solutions that cannot undergo any modification after administration [49]. The prepared hydrogel formulations (F7–F8) showed higher viscosity value than the corresponding ISG formulations. Both formulations are used to prolong drug release, reduce frequency of application, and reduce the systemic effect due to the nature of these formulations’ viscosities. Low viscosity before gelling has been reported to be suitable for ophthalmic application. A viscosity value up to 3500 cP (at 25 °C and 10 rpm) has been mentioned to be appropriate in terms of applying convenience [16]. Song et al. reported satisfactory viscosity values of 700 ± 85, 1120 ± 49, and 4300 ± 120 mPa s (at 20 rpm) at pH 5.5, 6, and 7, respectively, for carbopol/HPMC ocular ISG system [50].

#### 3.6.2. In Vitro Release Study

Previous reports indicated that the volume of a human’s tear is about 7 µL, and the cul-de-sac can accommodate a volume of fluid ~30 µL [51]. Accordingly, a known volume of the formulation and the STF in a ratio of 25:7 (formulation: STF) was used during this experiment.

All the studied ophthalmic preparations demonstrated an extended drug release over 6 h. The release of KET was higher from the trans-ethosomes nanovesicles compared with the same ophthalmic preparations containing the pure drug, as depicted in Figure 10. This behavior was mainly attributed to the smaller particle size of the developed trans-ethosomes when compared with the size of the pure drug suspension, which was expected to be in the form of coarse dispersion. Higher cumulative drug release percentage was obtained from F3 (101.65 ± 5.95%), F1 (71.12 ± 5.15%), F3 (66.71 ± 4.95%), F7 (63.49 ± 7.19%), F4 (59.78 ± 9.42%), F6 (50.89 ± 6.38%), F2 (46.48 ± 6.39%), and finally F8 (43.75 ± 5.38%). These results indicate a higher cumulative drug release from the carbopol-based formulation—the effect that could be attributed to easy diffusion of KET from this system.

Upon fitting the obtained data of the in vitro drug release to various kinetic models, all the studied formulations followed zero-order kinetics, as indicated by the highest value of the correlation coefficient (R) of the model, which was an indication that the release of KET was independent of the amount of drug released at different time points. The obtained results were in good agreement with previous work that reported zero-order release kinetics for itraconazole from polymeric micelles incorporated in situ ocular gel [52]. The (*n*) exponent obtained from the slope of the figure of log fraction of KET released (Mt/M∞) versus time demonstrated that the release mechanism follows anomalous (non-Fickian) transport (*n* > 0.45), which is a combination of both diffusion and erosion controlled-drug release.

It must be mentioned that biodegradability is one of the characteristics that makes the studied polymers beneficial for use in ophthalmic drug delivery [53]. Carbopol has been extensively used in ophthalmic formulation to improve the precorneal retention time of the applied drug. It provides the benefit of excellent mucoadhesive properties, as compared with other polymers, and high drug release. Its mucoadhesive properties is attributed to the interaction of the carbopol poly(acrylic acid) with the mucin by a combination of hydrogen bonding and electrostatic interaction [54]. Carbopol-based in situ gel demonstrated improving ophthalmic brinzolamide bioavailability and showed an extended drug release over a period of 8 h [55]. A good correlation between carbopol biodegradability and drug release has been previously mentioned. Incorporation of HPMC in the formulation decreases irritation and eye damage, which occurs due to the acidic nature of the prepared gel [56]. A combination of carbopol 940 (0.1% w/v), used as a gelling agent, and HPMC (0.4% w/v), used as a viscosity modifier, was used to develop an ISG system, which has demonstrated acceptable gel strength, sustained drug release over a period of 8h, no ocular toxicity or irritancy, in vivo elimination within 25 min, and effective suppression of inflammation for uveitis treatment [50]. Accordingly, the carbopol-based ISG formulation was selected for further investigation.

### 3.7. In Vivo Ocular Irritation Test

The eye irritation test on New Zealand white rabbits showed that the studied carbopol-based ISG formulations were nonirritants and could be tolerated. No macroscopic signs of swelling, redness, abnormal secretions, corneal opacity, congestion, hemorrhage, or marked destruction appeared in the iris. All animal eyes looked normal when compared with the control. Accordingly, the carbopol-based ISG formulations achieved a total score of zero in the irritation assessment using the classical Draize test and are considered as safe preparations for ocular administration. The same finding was previously reported for thermo-sensitive ISG formulation of pure KET [30]. Microscopic investigation of the corneal structure and integrity of the eye tissues will be discussed in the following section.

### 3.8. Fluorescence Laser Microscope Study

To examine the diffusion of the developed nanovesicles from the prepared carbopol-based ISG across the eye, the transport of these nanovesicles loaded with FITC from the anterior to the posterior part of the eye was studied. Figure 11 illustrates fluorescence laser microscopic images in the anterior part (cornea) and the posterior part (retina) of a rabbit’s eyes following treatment with the studied formulations. The fluorescence of the dye was clearly observed in the cornea of rabbits treated with the ISG formulation loaded with FTIC nanovesicles. The fluorescence was noted in the corneal samples at all the studied time points (after 1, 2, and 4 h). On the other hand, little fluorescence was observed in the cornea of the ISG loaded with pure FITC due to the limited distribution of the pure dye in the corneal cells. The retina of the animals treated with the ISG formulation loaded with nanovesicles containing FTIC demonstrated good distribution of the nanovesicles to the posterior part of the eye, as indicated by the presence of fluorescence in all the studied samples. Further, the fluorescence of the dye was hardly seen in the retina of the rabbits treated with ISG formulation loaded with pure FITC. It must be mentioned that FITC is of limited solubility in water (less than 0.1 mg/mL in water); thus, it was coarsely dispersed in the ISG matrix of the ISG formulation loaded with the pure dye. This effect may result in rapid washing of the dye by the tears and the eye-blinking action and, thus, account for the poor distribution of the pure dye in the corneal cells. The successful permeation of some of the pure dye particles to the retina may be attributed to the long contact of the ISG formulation with the eye.

The obtained results indicate successful delivery of the nanovesicles to the posterior eye segment and may account for this formulation’s effectiveness in the treatment of fungal retinitis. Our results are in good agreement with previous work that reported the presence of a significantly higher KET concentration in the aqueous and vitreous humor following administration of drug solid lipid nanoparticles (SLNs) when compared with a pure drug suspension. The authors mentioned that the prepared drug SLNs may indirectly establish their effectiveness in the treatment of fungal retinitis and attributed this effect to the fact that more than 70% of the fluid in the vitreous humor moves toward and exits through the retina [57]. Our work is more advantageous since the prepared nanoparticles were successfully delivered to the retina. Finally, the observed corneal structure and integrity were unaffected by the formulation treatment. No side effects were noted upon the incorporation of ethanol in the development of ophthalmic preparation loaded with trans-ethosomes nanovesicles. These results are in good agreement with those of the previous studies, which indicated the use of ethanol in ocular surface surgeries and in the treatment of different corneal diseases [13]. It is previously stated that the rabbit eye is more susceptible to irritant substances than the human eye [58]. Accordingly, the prepared carbopol-based ISG formulation loaded with the optimized KET could be considered a promising ophthalmic delivery system in the treatment of fungal infections.

## 4. Conclusions

Optimized KET trans-ethosomes nanovesicles were successfully developed utilizing the Draper–Lin small composite design. The optimized trans-ethosomes were spherical in shape and illustrated cationic nanosized flexible particles of high drug entrapment efficiency. The optimized nanovesicles showed superior antifungal activity against a standard strain of *Candida albicans* when compared with a drug suspension and a plain trans-ethosomes formulation. Different ophthalmic formulations loaded with the optimized trans-ethosomes were developed and showed prolonged drug release, no irritation on the eyes of the New Zealand white rabbits, and were able to permeate deep into the posterior eye segments. Thus, this ophthalmic KET delivery system could be used to enhance the drug’s ocular permeation, rapid eye clearance, and short elimination half-life in the eye; however, stable antifungal activities using fungal-infected mouse models and ocular pharmacokinetic studies are required.

## Figures and Tables

**Figure 1 pharmaceutics-13-00151-f001:**
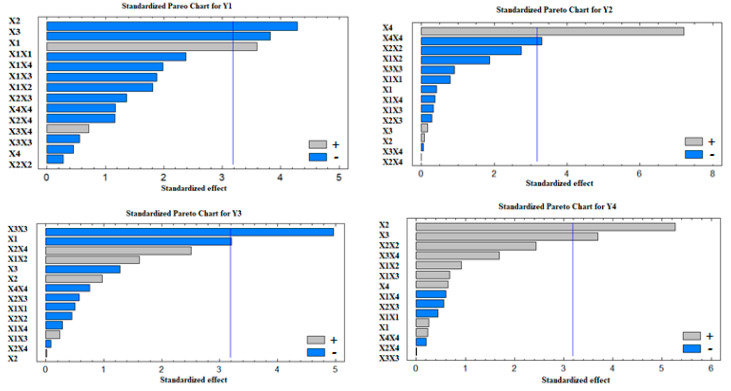
Standardized Pareto charts for the effect of the studied factors on Y_1_–Y_4_.

**Figure 2 pharmaceutics-13-00151-f002:**
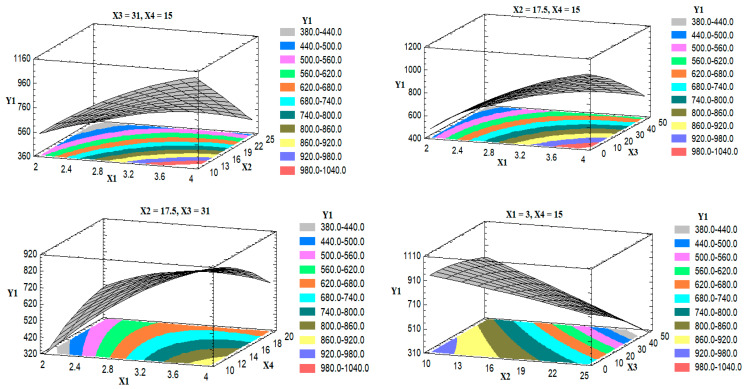
Estimated response surface plots for the effect of the studied factors on the particle size (Y_1_).

**Figure 3 pharmaceutics-13-00151-f003:**
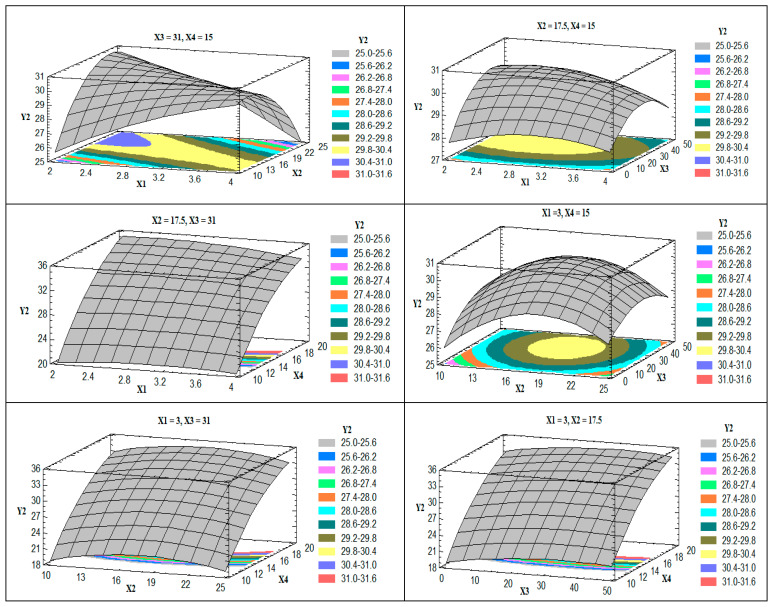
Estimated response surface plots for the effect of the studied factors on the zeta potential (Y_2_).

**Figure 4 pharmaceutics-13-00151-f004:**
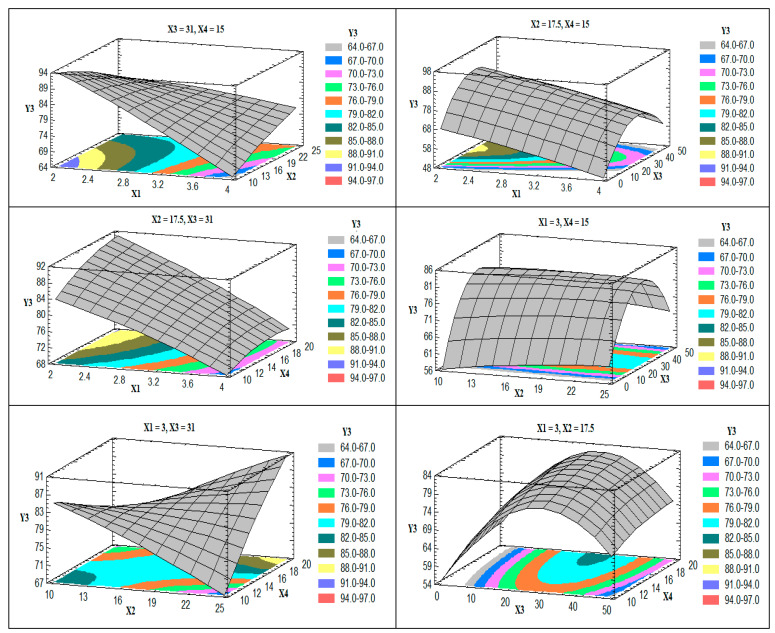
Estimated response surface plots for the effect of the studied factors on the entrapment efficiency (Y_3_).

**Figure 5 pharmaceutics-13-00151-f005:**
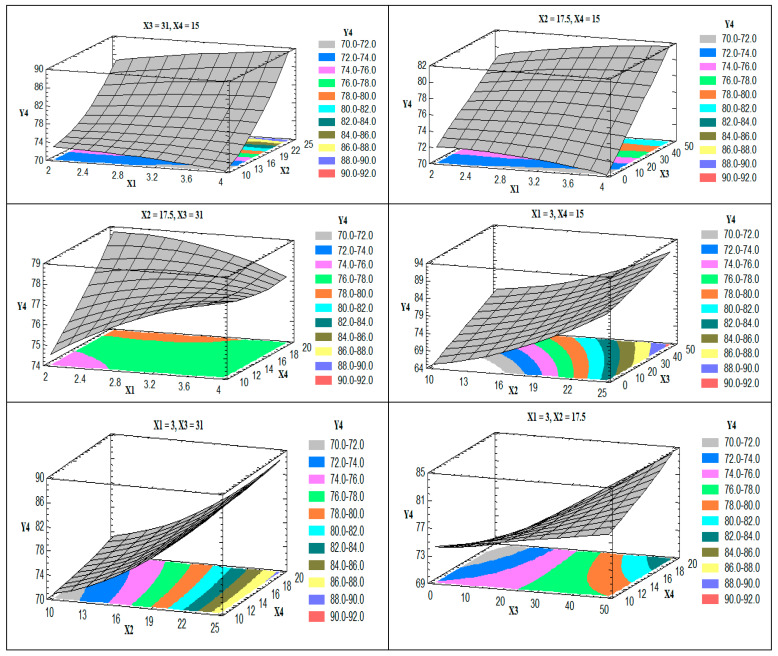
Estimated response surface plots for the effect of the studied factors on the flexibility (Y_4_).

**Figure 6 pharmaceutics-13-00151-f006:**
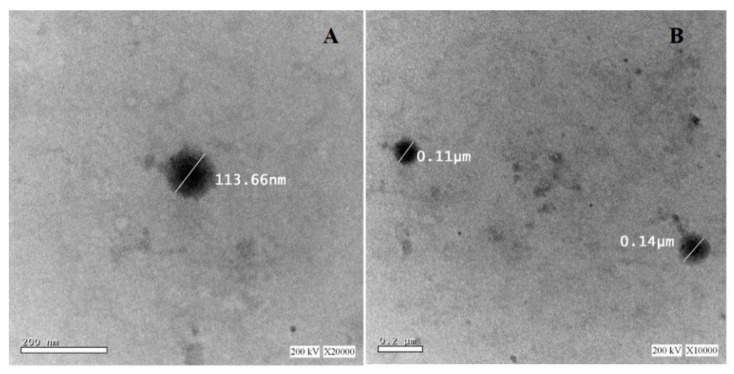
Transmission electron microscope for the optimized trans-ethosomal nanoparticles at 200 kV ×20,000 (**A**) and 200 kV ×10,000 (**B**).

**Figure 7 pharmaceutics-13-00151-f007:**
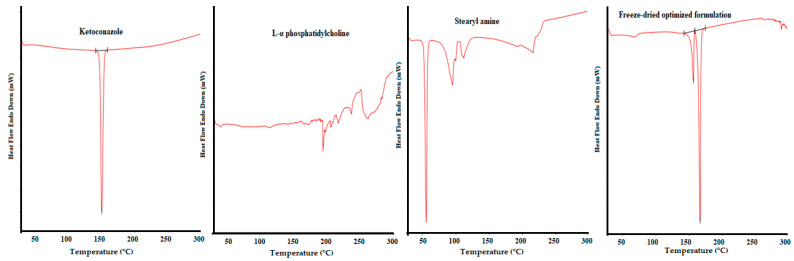
Differential scanning calorimetry thermograms of ketoconazole, phospholipid, stearyl amine, and the freeze-dried optimized formulation.

**Figure 8 pharmaceutics-13-00151-f008:**
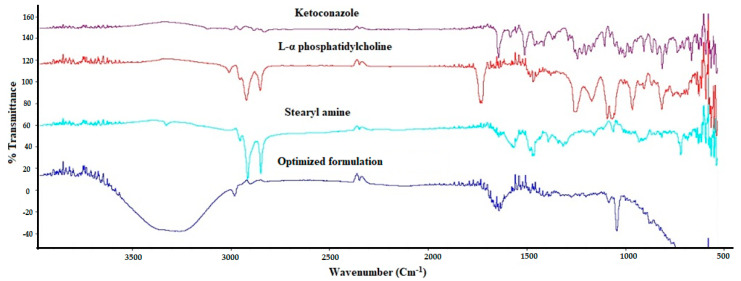
Fourier transform infrared spectra of ketoconazole, phospholipid, stearyl amine, and the freeze-dried optimized formulation.

**Figure 9 pharmaceutics-13-00151-f009:**
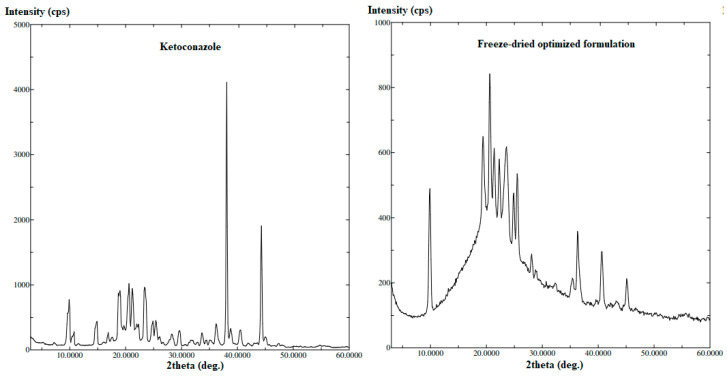
X-ray diffraction patterns of pure ketoconazole and the optimized freeze-dried trans-ethosomes formulation.

**Figure 10 pharmaceutics-13-00151-f010:**
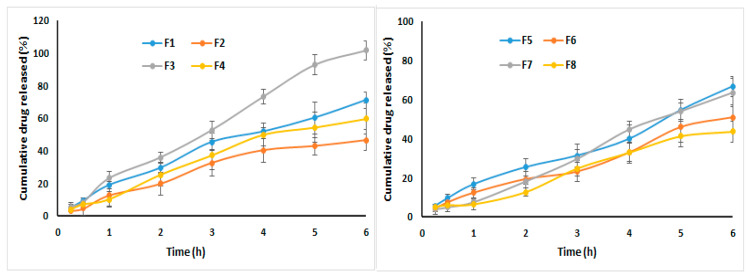
In vitro ketoconazole release from the prepared ophthalmic formulations.

**Figure 11 pharmaceutics-13-00151-f011:**
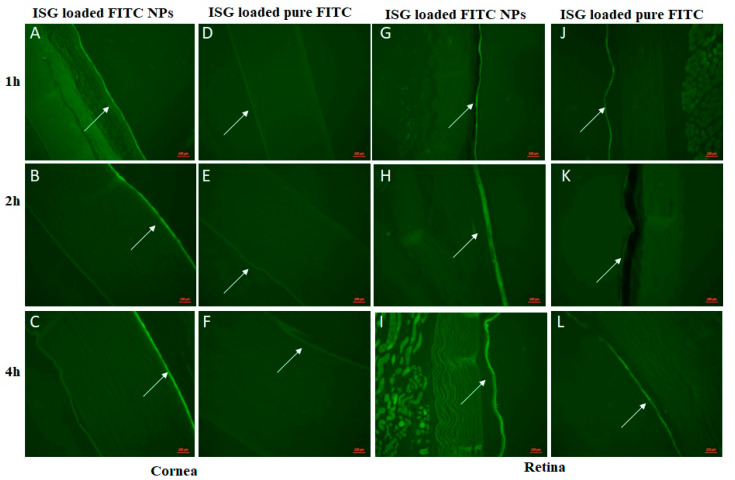
Fluorescence laser microscopic images of the eye’s sections on New Zealand white rabbits’ eyes following treatment with ISG formulation containing FITC-nanovesicles and ISG formulation loaded with pure FITC after 1, 2, and 4 h. Sections of the cornea (**A**–**F**) and retina (**G**–**L**) of rabbits’ eyes from both groups (arrows point to the fluorescent). Abbreviations: FITC, fluorescence isothiocyanate; ISG, in situ gel.

**Table 1 pharmaceutics-13-00151-t001:** Ketoconazole trans-ethosomes dependent and independent variables used in the Draper–Lin small composite experimental design.

Independent Variables	Level	
	Low	High
X_1_; Drug: phospholipid molar ratio	1:2	1:4
X_2_; % (w/w) of edge activator of the total lipid	10	25
X_3_; % (v/v) of ethanol in the hydration medium	12	50
X_4_; % (w/w) of stearyl amine of the total lipid	10	20
**Dependent Variables**	**Goal**	
Y_1_; Vesicle size (nm)	Minimize	
Y_2_; Zeta potential (mV)	Maximize	
Y_3_; Entrapment efficiency (%)	Maximize	
Y_4_; Flexibility (%)	Maximize	

**Table 2 pharmaceutics-13-00151-t002:** Experimental runs of ketoconazole trans-ethosomes nanovesicles and the observed values for the studied responses.

RUN	X1	X2	X3	X4	Y1	Y2	Y3	Y4
	MR	%	%	%	nm	mV	%	%
1	3	17.5	62.95	15	543.67 ± 2.08	29.80 ± 1.15	57 ± 4.16	80.76 ± 12.77
2	4.68	17.5	31	15	817.17 ± 20.74	29.03 ± 2.02	64.71 ± 2.20	75.15 ± 34.66
3	3	17.5	31	15	685.27 ± 85.33	29.03 ± 0.84	78.29 ± 3.60	77.96 ± 7.18
4	3	17.5	31	23.41	608.93 ± 54.51	36.80 ± 1.04	82.74 ± 3.20	78.13 ± 0.40
5	2	25	50	20	220.67 ± 28.08	34.67 ± 0.71	76.48 ± 14.39	92.68 ± 1.74
6	3	30.11	31	15	389.67 ± 48.50	26.07 ± 0.64	79.97 ± 0.27	96.24 ± 2.44
7	2	10	50	10	251.13 ± 18.02	15.03 ± 0.38	83.69 ± 4.83	73.34 ± 4.09
8	3	17.5	31	15	621.77 ± 6.60	28.83 ± 1.53	80.74 ± 0.81	80.05 ± 3.58
9	3	17.5	0	15	857.63 ± 151.99	29.13 ± 1.01	53.78 ± 48.53	70.93 ± 2.23
10	1.32	17.5	31	15	251.07 ± 114.12	30.37 ± 0.35	94.52 ± 2.84	73.83 ± 5.98
11	2	25	12	20	385.33 ± 72.71	34.60 ± 1.06	89.85 ± 2.85	83.48 ± 5.03
12	3	17.5	31	6.59	680.67 ± 11.06	12.87 ± 0.93	73.69 ± 5.21	74.87 ± 8.56
13	2	10	12	10	312.67 ± 52.08	13.83 ± 1.77	90.88 ± 2.51	69.85 ± 1.87
14	4	25	12	10	890.00 ± 94.30	15.17 ± 0.55	58.77 ± 1.97	89.30 ± 5.27
15	4	10	12	20	1061.00 ± 81.73	32.73 ± 0.67	48.04 ± 1.54	63.83 ± 0.44
16	4	10	50	20	817.00 ± 83.72	32.67 ± 0.91	42.27 ± 1.35	79.23 ± 0.67
17	4	25	50	10	317.63 ± 30.18	14.33 ± 0.76	48.44 ± 2.03	93.39 ± 3.28
18	3	4.89	31	15	1063.33 ± 131.96	25.77 ± 2.15	79.81 ± 9.21	69.69 ± 2.45

Abbreviations: X1, drug to phospholipid; X2, edge activator of the total lipid; X3, ethanol in the hydration medium; X4, stearyl amine of the total lipid; Y1, vesicle size; Y2, zeta potential; Y3, entrapment efficiency; Y4, flexibility; MR, molar ratio.

**Table 3 pharmaceutics-13-00151-t003:** Estimated effects of factors, *F*-ratio, and associated *p*-values for KET-TEs formulations particle size (Y_1_), zeta potential (Y_2_), entrapment efficiency (Y_3_) and flexibility (Y_4_).

Factor	Y_1_			Y_2_			Y_3_			Y_4_		
	Estimated Effect	*F*-Ratio	*p*-Value	Estimated Effect	*F*-Ratio	*p*-Value	Estimated Effect	*F*-Ratio	*p*-Value	Estimated Effect	F-Ratio	*p*-Value
X_1_	336.606	12.94	0.037 *	−0.797	0.160	0.713	−17.725	10.280	0.049 *	0.785	0.070	0.810
X_2_	−400.562	18.32	0.023 *	0.178	0.010	0.934	0.095	0	0.987	15.787	27.790	0.013 *
X_3_	−230.007	14.58	0.032 *	0.219	0.030	0.874	−4.576	1.650	0.289	7.134	13.700	0.034 *
X_4_	−42.656	0.21	0.679	14.229	52	0.005 *	5.381	0.950	0.402	1.938	0.420	0.564
X_1_X_1_	−154.389	5.66	0.098	−1.076	0.620	0.489	−1.927	0.250	0.649	−0.935	0.200	0.683
X_1_X_2_	−220.799	3.26	0.169	−4.849	3.540	0.157	11.666	2.610	0.205	3.603	0.850	0.425
X_1_X_3_	−147.542	3.52	0.185	−0.543	0.110	0.766	1.115	0.060	0.826	1.700	0.460	0.548
X_1_X_4_	−243.519	3.97	0.141	−0.949	0.140	0.737	−2.069	0.080	0.793	−2.36	0.360	0.589
X_2_X_2_	−18.356	0.08	0.957	−3.749	7.510	0.071	−1.733	0.200	0.682	5.057	5.930	0.093
X_2_X_3_	−107.872	1.88	0.796	−0.478	0.080	0.792	−2.685	0.330	0.604	−1.400	0.310	0.617
X_2_X_4_	−142.352	1.36	0.329	0.011	0	0.997	18.119	6.290	0.087	−0.815	0.040	0.848
X_3_X_3_	−36.634	0.32	0.612	−1.242	0.820	0.431	−19.057	24.710	0.016 *	0.0235	0	0.992
X_3_X_4_	56.313	0.51	0.526	−0.088	0	0.961	−0.405	0.010	0.936	4.255	2.850	0.189
X_4_X_4_	−76.126	1.38	0.326	−4.516	10.890	0.046 *	−2.917	0.580	0.502	0.486	0.050	0.830
R^2^	97.21%			98.52%			97.11%			97.24%		
Adj-R^2^	84.16%			91.64%			83.63%			84.37%		

Note: * Indicates significant effect of factors on individual responses, *p*-value < 0.05. Abbreviations: X_1_, drug to phospholipid; X_2_, edge activator of the total lipid; X_3_, ethanol in the hydration medium; X_4_, stearyl amine of the total lipid; X_1_X_2_, X_1_X_3_, X_1_X_4_, X_2_X_3_, X_2_X_4_, and X_3_X_4_ are the interaction terms between the factors; X_1_X_1_, X_2_X_2_, X_3_X_3_, and X_4_X_4_ are the quadratic terms of the factors; R^2^, R-squared; Adj-R^2^, adjusted R-squared; SEE, standard error of estimate; MAE, mean absolute error.

**Table 4 pharmaceutics-13-00151-t004:** Composition of the prepared ophthalmic in situ gels and hydrogels formulations and the obtained results for viscosity.

Formulation Code		Type of Polymers	Polymer Conc. (%)	Drug Form	Viscosity (cP) (25 °C, 30 rpm)	Viscosity (cP) After Gelation	Gelation Stimuli
**ISG**	F1	Poloxamer 407	16	KET-TNV	657 ± 49	17,863 ± 891	Temperature change (34 °C)
		HPMC	0.5				
	F2	Poloxamer 407	16	PD	477 ± 33	10,465 ± 680	Temperature change (34 °C)
		HPMC	0.5				
	F3	Carbopol 940	1	KET-TNV	967 ± 47	26,109 ± 1821	pH change (to 7.4)
		HPMC	0.5				
	F4	Carbopol 940	1	PD	853 ± 51	22,031 ± 1106	pH change (to 7.4)
		HPMC	0.5				
	F5	Sodium alginate	1	KET-TNV	1515 ± 81	39,390 ± 1771	Ionic gelation (CaOH)
		HPMC	0.5				
	F6	Sodium alginate	1	PD	1262 ± 82	31,550 ± 1557	Ionic gelation (CaOH)
		HPMC	0.5				
**Hydrogel**	F7	HPMC	2	KET-TNV	1470 ± 88	--	--
	F8	HPMC	2	PD	1361 ± 75	--	--

Abbreviations: ISG, in situ gel; KET-TNV, ketoconazole trans-ethosomes nanovesicles; PD, pure ketoconazole.

## Data Availability

Not applicable.

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
