# Peer review of "Study the Antifungal and Ocular Permeation of Ketoconazole from Ophthalmic Formulations Containing Trans-Ethosomes Nanoparticles"

_pharmaceutics, 2021, doi:10.3390/pharmaceutics13020151_

Round 1
Reviewer 1 Report
This study is to develop the ophthalmic formulation for sustained drug release of antifungal agent, KET.
In this study, in situ, gel formulation was prepared by establishing the optimal conditions for transe-ethosome nanoparticles, and the possibility of its ocular delivery system was investigated.
The acceptance of this study should be redetermined after considering the following;
1. English sentences need to be revised as a whole
- It is necessary to verity that trans-ethosome nanoparticles are homogenesously distributed within the hydrogel
- The mechanism of drug release from the nanoparticle within the hydrogel needs to be identified.
- The relationship between the biodegradation time of hydrogel or nanoparticle and drug release time needs to be revealed
- To identify the possibility as a ocular delivery system of KET, the antifungal activities of these formulations should be evaluated using fungal infectioned mouse model.
Author Response
Reviewer 1
The acceptance of this study should be redetermined after considering the following;
- English sentences need to be revised as a whole.
Reply:
The manuscript has been revised for grammar and typo mistakes.
- It is necessary to verity that trans-ethosome nanoparticles are homogenesously distributed within the hydrogel.
Reply:
During development of the in situ gel and hydrogel formulations, the specified weight of the studied polymer was added to known volume of the optimized drug loaded trans-ethosomes formulation over a magnetic stirrer. Stirring was continued until complete dispersion of the polymer and formation of homogenous mixture without lumps or precipitate. Also, HPMC was added as a viscosity modifier which prevent precipitation of the nanoparticles. This explanation has been added to the modified manuscript under the title “2.5.1. Preparation of in situ gel and hydrogel formulations”
- The mechanism of drug release from the nanoparticle within the hydrogel needs to be identified.
Reply:
Upon your request, the drug release kinetics and release mechanism were determined after fitting the in vitro release data to different mathematical models namely; Zero, First, Higuchi and Korsmeyer–Peppas. Results revealed that all the studied formulations followed zero order kinetics which indicates that the release of KET was independent of the amount of drug released at different time points. The release mechanism follows anomalous (non-Fickian) transport.
These data was added to the revised manuscript in the methodology section “2.5.2.2. In vitro release study” and the results and discussion section “4.8.2. In vitro release study”
- The relationship between the biodegradation time of hydrogel or nanoparticle and drug release time needs to be revealed.
Reply:
We completely agree with the reviewer point of view in that the biodegradation time is an important factor during the drug release process. Previous reports have indicated that Carbopol based in situ gel demonstrated prolonging pre-corneal residence time, improving ophthalmic brinzolamide bioavailability and showed an extended drug release over a period of 8 h. This behavior indicates good correlation between the polymer biodegradability and drug release. This explanation has been added to the revised manuscript in the results and discussion section “4.8.2. In vitro release study”.
- To identify the possibility as a ocular delivery system of KET, the antifungal activities of these formulations should be evaluated using fungal infectioned mouse model.
Reply:
We highly appreciate the reviwer comments. Unfortunayely we do not have the facilities to evaluate the fungal activity on infectioned mouse model and due to the Coronovirus pandemic it will be difficult to perform outside our Lab. This experiment will be submitted as a separate work along with the stability and ocular pharmacokinetic studies. This explanation has been added to the revised manuscript in the conclusion section.
Reviewer 2 Report
The manuscript entitled “Study the antifungal and ocular permeation of ketoconazole from ophthalmic formulations containing trans-ethosomes nanoparticles” is well designed and results are interesting. However, needs to address and justify the some of the major concerns.
- What is the toxicity of the ethanol on the ocular surface? Ethanol is not FDA approved for the preparation of ophthalmic formulations. Justify?
- Health professionals have stopped prescribing ketoconazole due to its high toxicity. Why it is used of ophthalmic.
Abstract:
- The abstract does not capture the central idea of the work well. It is better to summarize the experimental results according to the experimental sequence in the following part, and some important experimental results are not reflected in this part. Rewrite the abstract.
- Four formulation factors affecting the development of trans-ethosomal nanoparticles were optimized for their effects on the particle size, zeta potential, entrapment efficiency and nanoparticles flexibility. Rewrite the sentences. Very confusing.
- These are vesicular systems. Hence, use vesicle size, not particle size.
- Check for punctuation errors in the abstract.
Introduction:
The introduction of the manuscript is very poor. Write the recent developments in the keto formulation approaches. Also lacks proper references citation.
The authors are advised the following articles and book chapters in introduction for citing the latest available literature relevant to formulation approaches for keto research:
- Formulation, Optimization, and Evaluation of Ketoconazole Loaded Nanostructured Lipid Carrier Gel for Topical Delivery.
- Topical ketoconazole: a systematic review of current dermatological applications and future developments.
- Formulation and Characterization of Ketoconazole Loaded Nanosponges in Hydrogel for Treating Topical Fungal Infections.
- https://link.springer.com/article/10.1007/s12281-019-00338-6
- Ketoconazole has best suitable candidate for the oral delivery. What is the rationale for selecting keto and why the authors considered trans-ethosmes as delivery system? Explain and include in manuscript.
- Use keto abbreviation uniformly throughout the manuscript.
- Therefore, it is required to achieve a sufficient concentration of ketoconazole in the eye posterior segment for an effective treatment of fungal Keratitis. What is MIC of Keto for FK treatment.
Some polymers such as alginic acid, gellan gum, poloxamer and carbopol can be utilized to develop aqueous solutions that undergo gelation when instilled into the eye, resulting in prolongation of contact time, decreased frequency of the dose and increased transcorneal penetration. ISG formulations referred to polymeric solutions where there is a viscoelastic gel formation by the phase transition mechanism in response to physiological conditions. Cite following references for the statements.
https://www.mdpi.com/1999-4923/12/6/572; https://www.mdpi.com/2079-4991/9/1/33.
- Write the objective of the work very clearly.
Materials:
Delete SAJA. Write as kind gifted sample; Tween 80 use trade mark symbol.
Methods:
- What is the reason for application of CCD for optimization? What is the basis for selection of independent variables with their limits?
- Cite the references for CCD selection.
- What is drug loading in the formulations?
- Candida albicans ATCC 76615 should be italics.
- No reference method has been used or cited to assess antifungal activity. It is a requirement that the method be reproducible and comparable. One of the most suitable methods for evaluating antifungal activity is the CLSI microdilution method (CLSI M27-A3). https://www.sciencedirect.com/science/article/abs/pii/S0009308420300840.
- Further, diffusion methods are not as suitable for evaluating antifungal activity. However, you can adapt it to the agar well diffusion method. Explain.
- Please explain therapeutic window/safe concentrations of Ketoconazole in animals.
- Why different ISG formulations developed. Carbopol not used for the ISG formulation. it is act as mucoadhesive and forms gel like formulation with high concentration. How the authors confirm the formation of ISG with carbopol and other ISG agents.
- For in vitro release, write the temperature used.
- Write the animal ethical approval number and which guidelines followed for the in vivo irritation testing.
Results:
- Figure 8 is very poor quality and unable to read.
- Figure 10 – plate B is not good in shape and the formation of smear is not as per the records.
- Table 4, what is the ideal viscosity for the application of ISG formulations for ophthalmics.
- Figure 11 – merge both figures. Why the control formulation not used for the release studies.
- Delete figure 12 from the manuscript.
Author Response
Reviewer 2
The manuscript entitled “Study the antifungal and ocular permeation of ketoconazole from ophthalmic formulations containing trans-ethosomes nanoparticles” is well designed and results are interesting. However, needs to address and justify the some of the major concerns.
- What is the toxicity of the ethanol on the ocular surface? Ethanol is not FDA approved for the preparation of ophthalmic formulations. Justify?
Reply:
Ethanol has been reported to be widely used in ophthalmic surface surgeries such as photorefractive keratectomy or pterygium excision and to treat many corneal diseases such as recurrent corneal erosion and infectious keratitis. This finding has been mentioned in the introduction section, paragraph 3 based on previously mentioned study by Oh et al. For your references kindly referes to the following article
Oh JY, Yu JM, Ko JH. Analysis of ethanol effects on corneal epithelium. Investig Ophthalmol Vis Sci. 2013;54(6):3852–6.
Moreover, usually we apply 2-3 drops of the prepared formulation into the eye. Based on the optimized level of the studied factors used to develop the trans-ethosomes vesicles, the concentration of ethanol in the hydration medium will not exceeding 29%. Accoringly, minute amount of ethanol will be applied.
- Health professionals have stopped prescribing ketoconazole due to its high toxicity. Why it is used of ophthalmic.
Reply:
We completly agree with the reviwer comment in that oral ketoconazole pharmacotherapy may cause severe liver damages, adrenal gland problems and many other side effects. Our aim was to formulate ocular ketoconazole delivery system.
Ketoconazole is a lipophilic broad spectrum antifungal agent that has a high molecular weight of 531.44 Da. Accordingly, these characters hinder the drug transport across the corneal stroma during treatment of ophthalmic fungal infections especially those that affect the posterior segment.
This explanation has been illustrated in the modified manuscript in the introduction section, paragraph 2.
Abstract:
- The abstract does not capture the central idea of the work well. It is better to summarize the experimental results according to the experimental sequence in the following part, and some important experimental results are not reflected in this part. Rewrite the abstract.
Reply:
The abstract has been rewriten and more results have been added. Kindly refere to the abstract section.
- Four formulation factors affecting the development of trans-ethosomal nanoparticles were optimized for their effects on the particle size, zeta potential, entrapment efficiency and nanoparticles flexibility. Rewrite the sentences. Very confusing.
Reply:
The sentence has been rewritten in the modified manuscript. Kindly refere to the abstract section.
- These are vesicular systems. Hence, use vesicle size, not particle size.
Reply:
The term nanoparticles has been replaced with vesicles. Kindly refere to the abstract section.
- Check for punctuation errors in the abstract.
Reply:
Punctuation errors have been corrected. Kindly refere to the abstract section.
Introduction:
The introduction of the manuscript is very poor. Write the recent developments in the keto formulation approaches. Also lacks proper references citation.
The authors are advised the following articles and book chapters in introduction for citing the latest available literature relevant to formulation approaches for keto research:
- Formulation, Optimization, and Evaluation of Ketoconazole Loaded Nanostructured Lipid Carrier Gel for Topical Delivery.
- Topical ketoconazole: a systematic review of current dermatological applications and future developments.
- Formulation and Characterization of Ketoconazole Loaded Nanosponges in Hydrogel for Treating Topical Fungal Infections.
- https://link.springer.com/article/10.1007/s12281-019-00338-6
Reply:
Formulation approaches relevant to ketoconazole have been added to the revised manuscript. Kindly referes to the introduction section, Paragraph 2.
- Ketoconazole has best suitable candidate for the oral delivery. What is the rationale for selecting keto and why the authors considered trans-ethosmes as delivery system? Explain and include in manuscript.
Reply:
Due to the sever side effect encountered during oral ketoconazole treatment and the demand for an effective antifungal formulation used in the treatment of ocular fungal infections especially those affecting the posterior segment, ketoconazole which is a broad spectrum antifungal agent and trans-ethosomes which are effective nanocarrier in enhancing the drug permeation due to their unique components, were used.
This explanation has been added to the revised manuscript. Kindly referes to introduction section paragraph 3 and 5.
- Use keto abbreviation uniformly throughout the manuscript.
Reply:
KET has been used uniformly in the revised manuscript.
- Therefore, it is required to achieve a sufficient concentration of ketoconazole in the eye posterior segment for an effective treatment of fungal Keratitis. What is MIC of Keto for FK treatment.
Reply:
Previous study indicated that the minimum inhibitory concentration value of KET against yeast isolates obtained from cases of keratitis is 0.015-0.125 µg/ml. In this study, the optimized trans-ethosomes dispersion loaded with 0.1% w/v KET, that showed an entrapment efficiency of 94.97 ± 5.41%, was found to be an effective antifungal preparation. Also, the prepared trans-ethosomes dispersion has been developed into ophthalmic ISG formulation that was able to transport the drug loaded nanovesicles into the posterior eye segment. This explanation has been added to the revised manuscript in the results and discussion section under the title “3.7. Antifungal activity of KET trans-ethosomes nanoparticles”.
For your reference, this sentence has been modified to make it more clear to the reader. Please refers to the introduction section, paragraph 2.
Some polymers such as alginic acid, gellan gum, poloxamer and carbopol can be utilized to develop aqueous solutions that undergo gelation when instilled into the eye, resulting in prolongation of contact time, decreased frequency of the dose and increased transcorneal penetration. ISG formulations referred to polymeric solutions where there is a viscoelastic gel formation by the phase transition mechanism in response to physiological conditions. Cite following references for the statements.
https://www.mdpi.com/1999-4923/12/6/572; https://www.mdpi.com/2079-4991/9/1/33.
Reply:
Reefernces have been added to the revised manuscript. Kindly refers to the introduction section, paragraph 4.
- Write the objective of the work very clearly.
Reply:
The objective of the work has been rewritten.
Materials:
Delete SAJA. Write as kind gifted sample; Tween 80 use trade mark symbol.
Reply:
The required modifications have been addressed in the Materials section.
Methods:
- What is the reason for application of CCD for optimization? What is the basis for selection of independent variables with their limits?
Reply:
In this study, we have used Draper-Lin small composite experimental design to develop an optimized trans-ethosomal formulation. This tool is effective when we have to study the effect of four factors on a number of responses. Central composite design (CCD) is used when we have only three factors. Selection of the independent variables and their limits was based on our preliminary work and information found in the literature. This explanation has been added to the revised manuscript in the materials and methods section under the title “Draper-Lin small composite experimental design”.
- Cite the references for CCD selection.
Reply:
Draper-Lin small composite experimental design utilizing the StatGraphics Centurion XV version 15.2.05 software, StatPoint Technologies, Inc. (Warrenton, VA, USA) was used. The central composite design (CCD) was not selected as it is used when we have only three variables while in this work we investigated the effect of four variables using Draper-Lin small composite design.
- What is drug loading in the formulations?
Reply:
During development of the trans-ethosomes liquid dispersin, 0.1 % w/v based on the total formulation was used. The entrapment efficiency of the prepared optimized formulation was found to be 94.97%. Accordingly 100 mg was present in 100 ml of the optimized trans-ethosomal formulation of which 94.97 mg was entrapped in the vesicles.
- Candida albicans ATCC 76615 should be italics.
Reply:
The phrase “Candida albicans ATCC 76615” has been modified to the italic style.
- No reference method has been used or cited to assess antifungal activity. It is a requirement that the method be reproducible and comparable. One of the most suitable methods for evaluating antifungal activity is the CLSI microdilution method (CLSI M27-A3). https://www.sciencedirect.com/science/article/abs/pii/S0009308420300840.
Reply:
Reference for the method used to assess antifungal activity has been added to the revised manuscript. Kindly refers to the materials and methods section under the title “Antifungal activity of KET trans-ethosomes nanoparticles”.
- Further, diffusion methods are not as suitable for evaluating antifungal activity. However, you can adapt it to the agar well diffusion method. Explain.
Reply:
The agar well diffusion technique has been used to investigate the antifungal activity. This modification has been added to the revised manuscript in the materials and methods section under the title “Antifungal activity of KET trans-ethosomes nanoparticles”.
- Please explain therapeutic window/safe concentrations of Ketoconazole in animals.
Reply:
The therapeutic dose of ketoconazole in dogs is 10-40 mg/kg/day while, rats may be dosed up to 100 mg/kg/day. Since our formulation is intended for ophthalmic application which is characterized by low incidence of systemic side effects, no KET systemic adverse effects are expected from this formulation. This explanation has been added to the revised manuscript in the materials and methods section under the title “In vivo ocular irritation test”.
- Why different ISG formulations developed. Carbopol not used for the ISG formulation. it is act as mucoadhesive and forms gel like formulation with high concentration. How the authors confirm the formation of ISG with carbopol and other ISG agents.
Reply:
ISG system offers many advantages. It does not cause blurred vision or irritation upon administration. It provides extended drug residence time, an accurate dosing and increased shelf life.
Carbopol has been extensively used in development of in situ gelling system. A combination of carbopol 940, used as a gelling agent, and HPMC, used as viscosity modifier, was used to develop an ISG system that has showed acceptable gel strength, sustained drug release over a period of 8h, no ocular toxicity or irritancy, in vivo elimination within 25 minutes and effective suppression of inflammation for uveitis treatment. Accordingly, the carbopol based ISG formulation was selected for further investigation. This explanation has been added to the revised manuscript in the results and discussion section under the title “In vitro release study”.
- For in vitro release, write the temperature used.
Reply:
The temperature has been added in the revised manscript. Kindly refers to the materials and methods section under the title “In vitro release study”.
- Write the animal ethical approval number and which guidelines followed for the in vivo irritation testing.
- Reply:
Ethical approval number and guidelines for the in vivo irritation testing have been added to the revised manuscript. Kindly referes to the results and discussion section under the title “In vivo ocular irritation test”.
Results:
- Figure 8 is very poor quality and unable to read.
Reply:
Beeter quality figure has been added to the revised manuscript.
- Figure 10 – plate B is not good in shape and the formation of smear is not as per the records.
Reply:
Figure 10 has been removed and the zone of inhibition (mm) has been included in the revised manuscript.
- Table 4, what is the ideal viscosity for the application of ISG formulations for ophthalmics.
Reply:
Low viscosity before gelling has been reported to be suitable for ophthalmic application. A viscosity value up to 3500 cP (at 25 °C and 10 rpm) has been mentioned to be appropriate in term of applying convenience (16). Song et al reported satisfactory viscosity values of 700 ±85, 1120±49 and 4300±120 mPa s (at 20 rpm) at pH 5.5, 6 and 7, respectively for carbopol/HPMC ocular ISG system (50). This explanation has been added to the revised manuscript in the results and discussion section under the title “Rheological properties”
- Figure 11 – merge both figures. Why the control formulation not used for the release studies.
Reply:
Both figures have been merged.
The non-medicated carbopol based ISG formulation (control formulation) was used during the analysis as a blank. This explanation has been added to the revised manuscript in the materials and methods section under the title “In vitro release study”.
- Delete figure 12 from the manuscript.
Reply:
The required figure has been deleted.
Round 2
Reviewer 2 Report
The manuscript modified as per the reviewers comments. But, it needs major English language edits.
Author Response
The English language of the manuscript has been edited. An editing certificate has been attached to the cover letter.
